# L-Cysteine/Silver Nitrate/Iodate Anions System: Peculiarities of Supramolecular Gel Formation with and Without Visible-Light Exposure

**DOI:** 10.3390/gels10120809

**Published:** 2024-12-09

**Authors:** Dmitry V. Vishnevetskii, Elizaveta E. Polyakova, Yana V. Andrianova, Arif R. Mekhtiev, Alexandra I. Ivanova, Dmitry V. Averkin, Vladimir G. Alekseev, Alexey V. Bykov, Mikhail G. Sulman

**Affiliations:** 1Department of Physical Chemistry, Tver State University, Building 33, Zhelyabova Str., Tver 170100, Russia; elizabeth03pol@gmail.com (E.E.P.); nuri-chan-87@mail.ru (Y.V.A.); 2Institute of Biomedical Chemistry, 10 Building 8, Pogodinskaya Str., Moscow 191121, Russia; 3Department of Applied Physics, Tver State University, Building 33, Zhelyabova Str., Tver 170100, Russia; alex.ivanova33@yandex.ru; 4Russian Metrological Institute of Technical Physics and Radio Engineering, Worker’s Settlement Mendeleevo, Building 11, Moscow 141570, Russia; averkindmitry@gmail.com; 5Department of Inorganic and Analytical Chemistry, Tver State University, Building 33, Zhelyabova Str., Tver 170100, Russia; vg_alekseev@rambler.ru; 6Department of Biotechnology, Chemistry and Standardization, Tver State Technical University, A. Nikitina Str., Building 22, Tver 170026, Russia; bykovav@yandex.ru (A.V.B.); sulmanmikhail@yandex.ru (M.G.S.)

**Keywords:** l-cysteine, silver nitrate, iodate anions, low-molecular-weight gelators, self-assembly, photosensitivity, visible light, silver-containing nanoparticles, anion detection

## Abstract

In this study, novel anion photo-responsive supramolecular hydrogels based on cysteine–silver sol (CSS) and iodate anions (IO_3_^−^) were prepared. The peculiarities of the self-assembly process of gel formation in the dark and under visible-light exposure were studied using a complex of modern physico-chemical methods of analysis, including viscosimetry, UV spectroscopy, dynamic light scattering, electrophoretic light scattering, scanning electron microscopy, energy-dispersive X-ray spectroscopy, and X-ray photoelectron spectroscopy. In the dark phase, the formation of weak snot-like gels takes place in a quite narrow IO_3_^−^ ion concentration range. The visible-light exposure of these gels leads to an increase in their viscosity and dramatic change in their color. The morphology of gels alters after light irradiation that is reflected in the formation of a huge number of spherical/elliptical particles and the thickening of the fibers of the gel network. The interaction of CSS with IO_3_^−^ anions has features of a redox process, which leads to the formation of silver iodide/silver oxide nanoparticles inside and on the surface of CSS particles. CSS possesses selectivity only to IO_3_^−^ anions compared to many other inorganic ions relevant for humans and the environment. Thus, the CSS/IO_3_^−^ system is non-trivial and can be considered as a novel low-molecular-weight gelator with photosensitive properties, as another way to produce silver iodide nanoparticles, and as a new approach for IO_3_^−^ ion detection.

## 1. Introduction

Supramolecular gels formed by interactions between low-molecular-weight compounds or gelators (LMWGs) are a unique example of a gel’s state [1,2]. The non-covalent nature of chemical bonds plays a key role in the self-assembly process of LMWG-based gels [3]. This fact determines the potential of using such objects in various fields of technology and medicine [4]. In most cases, these systems include a single component [5]. However, multicomponent gels have also been actively explored recently [6]. Due to the supramolecular structure of a gel network, the gelation process is initiated by various stimuli: temperature, ultrasound, pH, specific chemical reactions, catalysts, electrolytes, and photochemical reactions [2,3,7,8,9,10].

Anion-responsive supramolecular LMWG-based gels have attracted a huge amount of attention from scientists because various anions play an important role not only in many biological processes but also in the environment [11,12,13]. To date, it is known that different low-molecular-weight anions added to initial gels mostly lead to gel-to-sol or gel-to-gel transitions [7,14]. Several scientific groups have shown some anions can cause a sol-to-gel transition; that is, they act as the initiators of gel formation [8,9,10,15,16,17]. This stimuli-responsive property opens up prospects of using these objects as sensors for various anions, for instance, as smart materials for pollutant removal [18].

Compared to internal stimuli, light is another important but external trigger for gel preparation. Photo-responsive LMWGs are an interesting example of self-assembly leading to supramolecular gel formation [19,20,21]. Organic compounds containing chromophore groups mostly act as photosensitive gelling agents. The mechanism of action of such systems consists of the following processes: isomerization, dimerization, bond formation, bond cleavage, and exciton formation. Photoisomerizable gelators can undergo a reversible sol–gel transition. They include azobenzenes [22], stilbenes [23], and alkenes [24]. Gel to sol transition, change in color, and electronic properties are typical features of ring-opening or ring-closing LMWGs, such as dithienylethenes [25], spiropyrans [26], and 2H-chromene [27]. Reactions on the basis of coumarins [28], anthracenes [29], and alkenes [30] tend to proceed via the photodimerization process, which also leads to the destruction of a gel network. Photopolymerization takes place for some LMWGs containing diacetylene or butadiene moieties [31]. As a result of this process, the gel strengthens. Finally, one useful photoreaction in chemistry is photocleavage. Unexpectedly, this reaction causes the sol-to-gel transition. LMWGs with a carboxy-2-nitrobenzyl group can partake in this process [32]. The abovementioned photo-responsive LMWGs have a variety of applications in the following biomedical fields: drug delivery/release, tissue engineering, 3D photopattern gels, and cell differentiation [33,34]. Another important application of these systems is as sensors, such as logic gates [35].

We recently reported the synthesis of supramolecular hydrogels on the basis of sulfur-containing amino acids and silver salts [15,36,37]. In most cases, the preparation of these systems includes two stages: the reaction of aqueous solutions of L-cysteine with silver nitrate leads to the formation of a colloidal solution (cysteine–silver sol or CSS) of cysteine–silver nanoparticles; and the addition of various low-molecular-weight inorganic anions initiates a sol–gel transition. This not only influences the morphology of a gel network, but also different properties of final gel systems [9,16,38,39,40,41]. Thus, the role of anions is crucial for this process.

This study relates to the discovery of an unusual anion photo-responsive gel system based on L-cysteine, silver nitrate, and iodate anions. It has been shown for the first time that iodate anions added to CSS can simultaneously act as a gelling and photosensitive agent in the visible region of wavelengths. Weak supramolecular gels were obtained in the dark by mixing CSS and KIO_3_ at a millimolar range of anion concentration. The exposure of visible ambient light to these gels strengthens them and dramatically changes their color. The dark phase of the process proceeds with a redox reaction, and the light one through the photoinduced decay of previously obtained products. CSS-based gels have photo-responsive selectivity only to iodate ions in comparison with many other anions found in the environment and consumed by humans with food and water. Thus, the obtained results can open avenues not only for the synthesis of novel anion photo-responsive LMWG-based gels, but as a new approach for iodate anion detection.

## 2. Results

### 2.1. Interaction of CSS with ClO_3_^−^, BrO_3_^−^, and IO_3_^−^ Anions

Recently, it has been discovered that only the fluoride anion among halide anions can form stable thixotropic gels with cysteine–silver hydrosol [9]. Therefore, it was of interest to investigate the behavior of CSS upon the addition of single-charge ions of corresponding halogen oxides. Based on our previous research, we expected either the precipitation of silver salts or the formation of hydrogels. The results were doubly unexpected (Figure 1). According to our earlier studies, CSS is a colloidal solution of nanoparticles with a core–shell structure and positive surface charge value [41]. To initiate the gelation process, it is necessary to reduce this charge value by adding various low-molecular-weight anions to CSS [9,37]. This study’s results showed that only the iodate anion leads to gel formation (Figure 1A). The process occurs at a fixed CSS concentration with varying IO_3_^−^ ion content in the millimolar range (Figure 1B). Yellowish precipitates occur at anion concentrations above 2.5 mM. Additionally, we observed that exposing these weak snot-like gels to visible-light changes their color from greenish-yellow to increasingly brown, strengthening the gel structure. The resulting gels exhibit reversible gel–sol transitions, demonstrating their thixotropic property. The color change ceases completely when the gels are moved to a dark place. No gel formation or color change was observed for ClO_3_^−^ and BrO_3_^−^ anions in the same concentration range (Figure 1B). Therefore, we focused on investigating iodate-based gels in more detail using a complex of physico-chemical methods of analysis. It is worth noting that there is no information in the literature about similar behavior in any system containing IO_3_^−^ ions.

### 2.2. Self-Assembly Process Study in CSS/IO_3_^−^ System

It is evident that exposure to visible light leads to the restructuring of a gel network. Therefore, it is important to monitor these changes by measuring the viscosity of the systems. The visual assessment of the viscosity of gels was conducted by rotating test tubes by 180 degrees (Figure 1A). Each state was assigned an integer from 0 to 5 [15]. As a result, diagrams showing the strength of gels depending on the iodate ion content were generated (Figure 2a,b). It can be observed that in a dark environment, starting from a certain concentration of IO_3_^−^ ions, the viscosity of gels reaches a point of 3 and remains constant (Figure 2a). After exposing these gels to visible light, the graph shows a dome-shaped dependence (Figure 2b). The strongest gels (5 points) are obtained within a narrow concentration range of iodate ions, as observed for other anions [9,15,16]. Previous studies using vibrational viscosimetry have shown that for gels with 5 points, the higher the concentration of fluoride ions, the lower the gel viscosity [9]. A similar trend was observed for the iodate anion, regardless of the gel preparation conditions (Figure 2c,d). However, after irradiation, the viscosity of gels at the initial time moment is higher (Figure 2d) compared to non-irradiated gels (Figure 2c), and these values remain relatively stable over time, indicating the completion of the gel network structuring processes. These results support the findings of the visual experiment. To further understand the observed phenomena at the macro-level, it is necessary to study the systems at the micro- and nano-levels.

The gel network structure of the systems was examined by SEM (Figure 3). The morphology of gels forming before and after their visible-light exposure is elongated spider web-like fibers. A similar gel network structure was observed for CSS/two-charged anions (SO_4_^2−^ or MoO_4_^2−^) systems [37,38]. However, one can clearly see that moving from non-irradiated to irradiated samples, firstly, the density of the gel carcass becomes higher, secondly, a large number of spherical/elliptical particles are formed, and somewhere, gel network fibers thicken significantly. An EDS analysis shows the presence of all initial components in the gel network structure. The obtained results are in good agreement with the discussed data of gel strength diagrams and measurements of their viscosity. One can assume that such morphological changes are responsible not only for gel reinforcement but also for altering their color.

The UV analysis revealed significant changes in the absorption spectra of gels based on iodate ions compared to the spectrum of the initial CSS (Figure 4). The position of the main absorption bands of CSS and their intensity at 310 and 390 nm remained constant after visible-light exposure of the hydrosol (Figure 4a,b, graph 4). There were no changes in the spectra of the initial electrolyte solutions (Figure 4,a,b, graphs 1, 2, 3) as well as the CSS/ClO_3_^−^ and CSS/BrO_3_^−^ systems (Figure 4,a,b graphs 5). The addition of IO_3_^−^ anions to the CSS resulted in the appearance of new absorption peaks at 350–360 and 425 nm after 24 h of samples being kept in a dark place (Figure 4a). The higher the concentration of the iodate ion, the stronger the absorption of these bands. The absorbance and position of these bands did not change after one week of samples being kept in the dark. After the irradiation of CSS/IO_3_^−^ systems, the position of these two peaks remained unchanged, but the absorption at 425 nm increased, especially noticeable in the visible wavelength range from 450 to 750 nm (Figure 4c). The kinetics of the observed phenomenon showed that, regardless of whether the samples were in the dark or in the light, the same accumulation of the product occurred at 350–360 nm after an equal period of time (Figure 4d). The difference was in the higher absorption in the spectral range from 450 to 750 nm of the irradiated system compared to the one in a dark. Thus, the reaction between iodate anions and CSS was initiated in the absence of light, then paused at some point and further continued only under visible-light exposure.

Figure 5 demonstrates the results of the measurement of sizes and surface zeta potentials of formed particles, which are elemental units of the resulting gel network. The particle size distribution is unimodal (Figure 5a,b). The average particle sizes in pre-irradiated systems are about 100–200 nm, and they change slightly after visible-light exposure. The polydispersity coefficients practically do not alter. The values of the zeta potential of particles in non-irradiated systems decrease (Figure 5c) in comparison with the initial CSS (+60–70 mV), which is typical for such systems [9,15,37,38]. After light exposure, a single dependence is observed: the values of the zeta potential grow and return almost to the values of the initial CSS (Figure 5d). Thereby, IO_3_^−^ anions interact with the positively charged surface of CSS particles, lowering its charge, and then they probably undergo some transformations that are especially noticeable after visible-light exposure.

XPS analysis, a powerful technique for investigating the composition and chemical state of surface components in various compounds, was performed to clarify the observed phenomena (Figure 6 and Figure 7). As shown in Figure 6 the synthesized gels (in the form of xerogels) contain oxygen, carbon, iodine, silver, sulfur, and nitrogen on their surface and in near-surface atomic grids in appropriate concentrations.

As shown in Figure 7a, the peaks of photoelectron sublevels of the iodine atom are observed at 619.5 and 631 eV. The former corresponds to iodide anions (I^−^) probably in the silver iodide (AgI) state [42]. The binding energy of Ag 3 d_5/2_ at 368.5 eV (Figure 7b) and the kinetic energy of the M_5_VV component of the Auger-series of silver at 350.7 eV (Figure 7c) in accordance with the Wagner diagram, indicate that this element is present in one chemical state as silver oxide (Ag_2_O) [43]. These data also suggest the possibility of the formation of a mixed phase of AgI/Ag_2_O [44]. Sulfur is in two oxidation states (Figure 7d): S (−2) at 162.3 eV and S (+6) at 168.2 eV (probably as -SO_3_H or SO_4_^2−^) [45]. The O 1s line (Figure 7e) has a complex composition due to the superposition of chemical oxygen states corresponding to oxygen in sulfates (531.5 eV) and carboxyl groups (532.2 and 533.8 eV), as well as for two oxygen states obviously bonded with silver (531.2 and 532.0 eV) [43]. The typical state of O (−2) in oxides usually corresponds to binding energies of 530–528 eV, higher values of 532–530 eV typically related to OH groups and oxygen in oxygen-containing anions such as sulfate, carbonate, phosphate anion, etc. Therefore, the binding energy at 531.2 eV seemed to correspond to the mixed phase of AgI/Ag_2_O or OH-groups on the surface of the silver oxide, and the energy of 532 eV may be connected with the chemically sorbed water. Nitrogen can be attributed to non-protonated (399.6 eV) and protonated (401.6 eV) amino groups (Figure 7f). Thus, the iodate anions do undergo a chain of transformations.

### 2.3. Interaction of the CSS with Various Anions

Due to the CSS being more sensitive to the IO_3_^−^ anion compared to ClO_3_^−^ and BrO_3_^−^ anions, we decided to test the effect of other anions. We selected anions commonly consumed by humans through food and water (Figure 8). These included oxidizing, reducing, and inert agents. Upon immediate mixing of hydrosol with the corresponding electrolyte, no visible changes in the colloidal stability of the system or its color were observed (Figure 8a). After one day in a dark environment, systems with halogens, except for fluoride ion, began to opalesce (Figure 8b), as previously observed in our recent study [9]. A slight color change was noticeable in the system with iodate anion. Further exposure to visible light for an hour resulted in gradual color intensification only in the iodate anion-based sample (Figure 8c–e). UV spectroscopy data revealed a distinctive characteristic spectrum only in the iodate anion-based system (Figure 8f).

## 3. Discussion

Based on the results obtained, the IO_3_^−^ anion added to CSS serves multiple purposes: it initiates the gelation process and imparts photosensitivity to the system. The latter property is quite significant and non-trivial for the system being studied. The color change in the system may be attributed to the formation of a complex compound or the progression of a redox reaction. Complexes containing the iodate anion are typically white or colorless [46]. It is well known that iodates are strong oxidizing agents, and therefore, other components of the cysteine–silver sol must act as reducing agents. The CSS comprises nanoparticles of a core–shell structure [37,38,41]. The core consists of zero-valent silver nanoparticles (AgNPs), while the shell is constructed from cysteine–silver complexes (CYS/Ag^+^) with characteristic absorption bands at 390 nm (surface plasmon resonance—SPR) and 310 nm (argentophilic interactions), respectively (Figure 4). Amino and carboxyl groups are located on the particle surfaces. The hydrosol has a pH below 3, and the particles are in the zwitter-ionic form with an isoelectric point of 5.5 [47]. Therefore, sulfur or silver atoms in CSS structure can potentially act as electron donors for IO_3_^−^ anions, leading to a decrease in the oxidation state of I (+5) to I (−1). XPS analysis confirmed this phenomenon (Figure 7a), as typical binding energies for the IO_3_^−^ anion (623.5 and 634.9 eV) [42] decreased. Additionally, sulfur and silver atoms exhibited an increase in oxidation state (Figure 7b–d). In acidic media, the interaction of I^−^ with IO_3_^−^ results in the formation of I_2_ molecules. The aqueous solution of molecular iodine has a yellowish to brown color depending on its concentration. Although we conducted several qualitative reactions with starch, thiosulfate anion, and chloroform extraction, the color and UV spectra of the system remained unchanged. The XPS analysis was unable to detect molecular iodine due to its evaporation during the experiment. Iodine aqueous solution exhibits characteristic peaks in UV spectra at 350–360 and 450–460 nm, corresponding to I_3_^−^anion and I_2_, respectively [48]. CSS/IO_3_^−^ systems displayed absorption bands at 350–360 nm (plateau) and 425 nm (Figure 4, graphs 6). It is plausible that the former band is associated with the triiodide anion (I_3_^−^) formed by the interaction between I^−^ and I_2_, potentially bonded with silver atoms and protonated amino-groups of CSS because the starch reaction did not result in color change. The band at 425 nm, with a shoulder in the longwave region (Figure 4c), may be related to the SPR of AgNPs and various silver-containing nanoparticles [49,50]. According to XPS, this band corresponds to AgI/Ag_2_O NPs and likely contributes to the brown color of the formed gels. The reaction kinetics (Figure 4d) indicates a simultaneous decrease in absorbance at 310 nm (Ag(I)-Ag(I) interactions in L-cysteine/Ag^+^ complexes) and a concomitant increase in absorbance at 350–360 nm. Combining XPS and UV results, it can be inferred that I^−^ and I_3_^−^ anions disrupt these argentophilic interactions forming AgI phase (425 nm) and Ag-I_3_^−^ complexes (350–360 nm). The Ag_2_O phase (425 nm) is likely obtained through the direct oxidation of AgNPs core of CSS by IO_3_^−^ anions. Therefore, the redox reaction in the CSS/IO_3_^−^ system occurs without light, but after exposure to visible light, certain structures (adducts) may begin to decompose. Banin and Ruhman reported the ultrafast (4–5 ps) photodissociation of I_3_^−^ into I_2_^−^ in ethanol solution [51]. Kühne and Vöhringer later confirmed these findings [52]. In the CSS/IO_3_^−^ system, I_2_^−^ can interact with Ag(I) in the CSS structure to form AgI phase and I^−^. AgI is also unstable under light irradiation [53]. Consequently, the Ag_2_O phase likely forms on the surface of AgNPs core, while the AgI phase is generated in the shell (CYS/Ag^+^) and on the surface of CSS nanoparticles. These AgI/Ag_2_O particles are visible in the SEM images (Figure 3). The AgI/Ag_2_O NPs strengthen the gel network, leading to an increase in gel viscosity (Figure 2).

On one hand, iodine deficiency makes IO_3_^−^-containing substances a crucial source of molecular iodine for humans. However, an excess of iodine or iodide can lead to goiter and hypothyroidism [54]. Quality control of iodine-containing products is essential. Among various analytical methods for detecting iodate anions [55], spectrophotometry is the quickest, easiest, and most cost-effective method. Based on our results (Figure 8) and the existing literature, CSS can be considered a novel reagent for detecting IO_3_^−^ anions. On the other hand, the CSS/IO_3_^−^ system presents a new approach for creating composite AgI NPs. While the literature is limited on AgI NPs synthesis, these nanoparticles have diverse applications in fields like photocatalysis, sensors, fast ionic conductors, and antibacterial therapy [50,56]. Lastly, the CSS/IO_3_^−^ system represents the first example of silver-iodine-containing supramolecular hydrogels with anion-photo-responsive properties.

## 4. Conclusions

In conclusion, novel supramolecular hydrogels based on low-molecular-weight compounds such as L-cysteine, silver nitrate, and iodate anions have been synthesized for the first time. The snot-like gel formation was observed in the dark in a range of IO_3_^−^ ions concentration from 0.826 to 1.67 mM. The exposure of these gels to visible light for 1 h strengthened them, as reflected in their increased viscosity. During irradiation, the color of the gels turned from greenish-yellow to brown. The dark and light phase processes were related to the formation of silver iodide/silver oxide nanoparticles. It was found that the initial cysteine–silver hydrosol was sensitive only to iodate anions (especially while exposed to visible light) compared to chlorate, bromate, halides, phosphates, carbonates, sulfates, nitrate, and nitrite anions. Future investigations will focus on deciphering the self-assembly mechanism in these systems and understanding their various practical applications. Furthermore, the possibility of detecting periodate anions and other strong oxidizing agents (e.g., MnO_4_^−^) will be explored.

## 5. Materials and Methods

### 5.1. Reagents

All reagents were used as received (Table 1). The systems under study were prepared using de-ionized water.

### 5.2. Synthetic Protocol for Sample Preparation

An amount of 2 mL of the initial hydrosol (CSS) was prepared according to our previous work [37]: an empty vessel was filled with 1.25 mL of an aqueous solution of L-cysteine (4.8 mM) followed by the addition of 0.75 mL of an aqueous solution of silver nitrate (10 mM). The ratio of amino acid to silver salt was 1:1.25. The resulting slightly cloudy mixture with a white-yellow shade was stirred for 1 min at room temperature (25 °C) and left for storage in a dark place for 3 h. A transparent colloidal solution (CSS) with a greenish-yellow color was formed.

To study the self-assembly process of CSS with electrolytes (KClO_3_, KBrO_3_, KIO_3_), different amounts of the potassium salt dissolved in water (10 mM) were added to 1 mL of hydrosol and stirred at room temperature (25 °C) for 1 min (Figure 1). The obtained solutions were placed in the dark for 24 h. After that, these systems were exposed to visible light for 1 h.

To examine the kinetics of UV spectra changing of the CSS/IO_3_^−^ system 20 (see Figure 1B, Table) two identical samples were prepared: 0.2 mL of KIO_3_ aqueous solution (0.01 M) was added to 1 mL of CSS. One of them was immediately put in the dark for 100 min, and the second one was exposed to visible light for 100 min.

To investigate the interaction of various electrolytes (Table 1) with CSS, 0.32 mL of the corresponding salt (10 mM) was added to 2 mL of hydrosol (Figure 8). After 24 h of staying in a dark place, samples were irradiated with visible light for 1 h.

Viscosimetry, SEM, EDS, DLS, ELS, and XPS were conducted only for CSS/IO_3_^−^-based hydrogels (systems 8, 10, 12, 14, and 16; see Figure 1B, Table) before and after exposure to visible light.

### 5.3. Viscosimetry

The SV-10 (A&D, Tokyo, Japan) vibratory viscometer was used to measure the apparent viscosity values of hydrogels. The sensor plates vibrated at a frequency of 30 Hz with a constant amplitude of about 1 mm. The samples (10 mL) were prepared in special polycarbonate cups (A&D, Tokyo, Japan) and then transferred to the viscometer, covered tightly with a black film to protect from light, and measurements were recorded for 10 min at 25 °C.

### 5.4. Scanning Electron Microscopy and Energy-Dispersive X-Ray Spectroscopy

A JEOL 6610 LV electron microscope (JEOL Ltd., Tokyo, Japan) with the Oxford INCA Energy 350 X-ray energy-dispersive microanalysis system (JEOL Ltd., Tokyo, Japan) was used to investigate the surface microstructure and elemental composition of gels. Scanning of the sample surface was conducted with an accelerating voltage of 15 kV in high-vacuum mode. Signals from low-energy secondary electrons responsible for topographical contrast, and high-energy backscattered electrons defining phase contrast and composition were used for image acquisition. X-ray spectral microanalysis involved registering and analyzing the energy spectra of characteristic X-ray radiation excited by electrons passing through the sample to determine the elemental chemical composition. Sample preparation involved spraying the samples onto a thin conductive layer of platinum surface and drying them in a vacuum (10^−4^ Pa). The average platinum coating time was 5 min.

### 5.5. UV Spectroscopy

The electronic spectra of the samples were recorded using a UV spectrophotometer Evolution Array (Thermo Scientific, Waltham, MA, USA) in a quartz cell with an optical path length of 1 mm.

### 5.6. Dynamic Light Scattering

The size of particles formed in the systems under investigation, as well as their zeta potential, was measured using Zetasizer Nano ZS (Malvern, Worcestershire, UK) equipped with a He-Ne laser (633 nm) with a power of 4 mW. The hydrogels were transferred into a sol state by shaking and then diluted two, four, and eight times. All measurements were carried out at 25 °C in the backscattering configuration at an angle of 173°, which provides the highest sensitivity of the device. The mathematical processing of the obtained results was performed using Zetasizer Software. The cross-correlation function *g*_2_(*τ*) was obtained:$$g_{2}\left(\tau \right)=\frac{\langle I(t)\cdot I(t+\tau )\rangle }{\langle I^{2}\rangle }$$

The equation showing the dependence of the *g*_2_(*τ*) function on the diffusion coefficient *D* has the following form:$$g_{2}\left(\tau \right)=1+C{\left[\int_{D_{min}}^{D_{max}}Z\left(D\right)\exp\left({-q}^{2}D\tau \right)dD\right]}^{2}$$

*Z*(*D*) is the distribution function of scattering particles based on the diffusion coefficients. This equation was solved using the cumulant method, resulting in the determination of the *Z*(*D*) function. The average *D* was calculated from the attenuation time, which is the time when the function decreases by a factor of e (2.74) times the cross-correlation function. The average hydrodynamic radius of the scattering particles was then calculated from the diffusion coefficient using the Stokes–Einstein formula D = kT/6πηR, where D represents the diffusion coefficient, k is the Boltzmann constant, T is the absolute temperature, η is the viscosity of the medium, and R is the radius of the scattering particles.

### 5.7. Electrophoretic Light Scattering

Electrophoretic mobility measurements of particles in samples were carried out in U-shaped capillary cuvettes. The hydrogels were transferred into a sol state by shaking and then diluted two, four, and eight times. Distributions of the zeta potential were calculated using the Henry equation UE = 2ezf(Ka)/3Z, where UE represents electrophoretic mobility, z stands for zeta potential, e represents dielectric constant, Z stands for viscosity, and f (Ka) represents Henry’s function; f (Ka) is equal to 1.5 for aqueous media.

### 5.8. XPS

The hydrogels were placed in a freezer at −20 °C for 24 h. The samples were then thawed at room temperature. The precipitates were centrifuged at 12 × 10^3^ rpm for 20 min. The supernatants were decanted, and the precipitates were dried to a constant weight in an oven at 40 °C in the dark.

XPS data were collected using Mg Kα (hν = 1253.6 eV) radiation with an ES-2403 spectrometer (Institute for Analytic Instrumentation of RAS, Saint Petersburg, Russia) equipped with an energy analyzer PHOIBOS 100-MCD5 (SPECS, Berlin, Germany) and X-ray source XR-50 (SPECS, Berlin, Germany). All data were acquired at an X-ray power of 250 W. Survey spectra were recorded at an energy step of 0.5 eV with an analyzer pass energy of 40 eV. High-resolution spectra were recorded at an energy step of 0.05 eV with an analyzer pass energy of 7 eV, corresponding to a half-width at half height of Ag 3d_5/2_. The auger-series of silver was recorded in 0.2 eV increments with the same analyzer settings. Samples were outgassed for 180 min before analysis and remained stable during the examination. Data analysis was performed using CasaXPS 2.3.24, SpecsLab 2.35. Binding energies (BEs) were determined with an error ± 0.1 eV.

## Figures and Tables

**Figure 1 gels-10-00809-f001:**
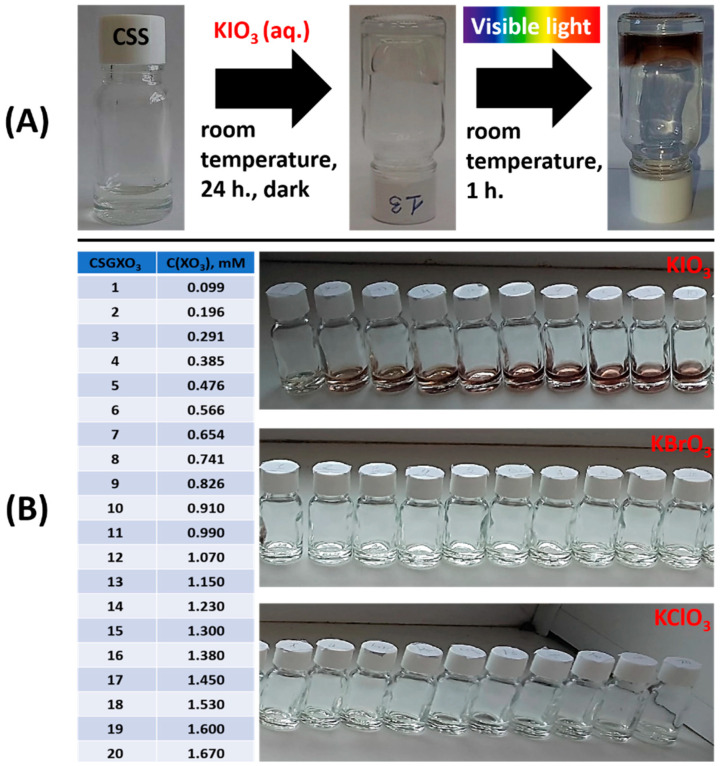
(**A**) Hydrogel formation in the CSS/IO_3_^−^ system. (**B**) The anion (XO_3_—ClO_3_, BrO_3_ or IO_3_) content in the systems (Table); photo of CSGXO_3_ systems 2, 4, 6, 8, 10, 12, 14, 16, 18, 20 (see Table, Figure 1B) after visible-light exposure for 1 h. CSGXO_3_—cysteine silver gels based on XO_3_^−^ anions.

**Figure 2 gels-10-00809-f002:**
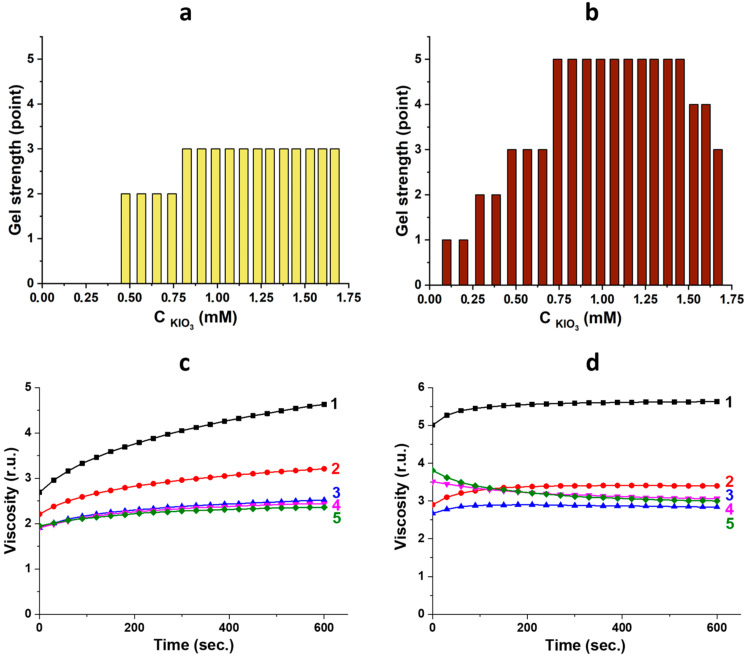
Concentration diagrams and viscosity dependence over time for CSS/IO_3_^−^-based gels before (**a**,**c**) and after (**b**,**d**) visible-light exposure for 1 h. Numbers 1, 2, 3, 4, and 5 on graphs (**c**,**d**) correspond to samples 8, 10, 12, 14, and 16 (see Table, Figure 1B).

**Figure 3 gels-10-00809-f003:**
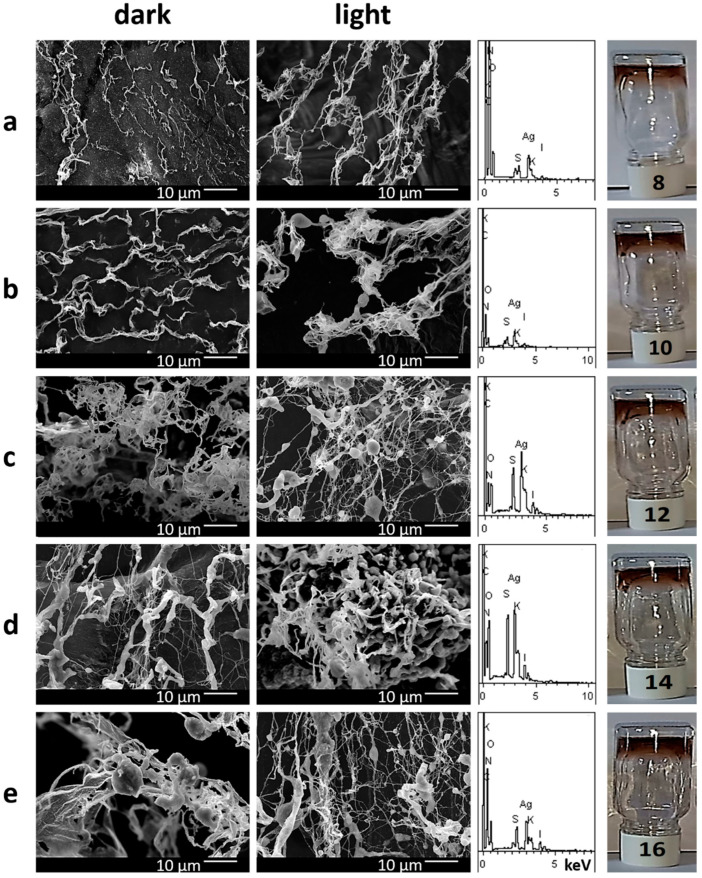
SEM images and EDS of hydrogels before (dark) and after visible-light exposure (light) for 1 h. Images (**a**–**e**) correspond to gel systems 8, 10, 12, 14, and 16 (see Table, Figure 1B). EDS data are presented for irradiated samples. Photos are provided for the corresponding irradiated gels.

**Figure 4 gels-10-00809-f004:**
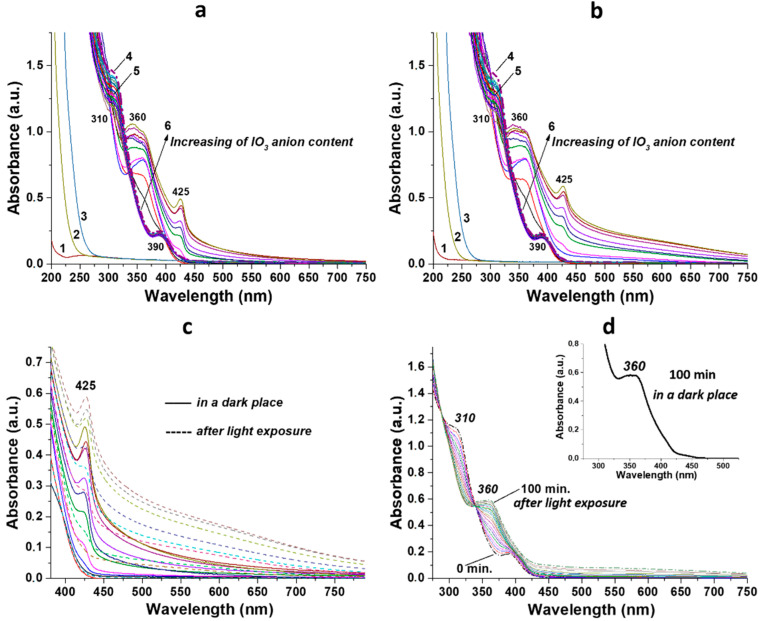
UV spectra of the systems before (**a**) and after (**b**) visible-light exposure: 1, 2, and 3—aqueous solutions of KClO_3_, KBrO_3_, and KIO_3_ respectively; 4—CSS; 5—CSS/ClO_3_^−^ and CSS/BrO_3_^−^; 6—CSS/IO_3_^−^. Data are presented for systems 2, 4, 6, 8, 10, 12, 14, 16, 18, and 20 (Figure 1B). (**c**) Comparison of UV spectra of the CSS/IO_3_^−^ systems before (bold lines) and after (dash lines) light irradiation. (**d**) Kinetics of UV spectra evolution for the CSS/IO_3_^−^ system 20 (Figure 1B) in the dark and under light exposure.

**Figure 5 gels-10-00809-f005:**
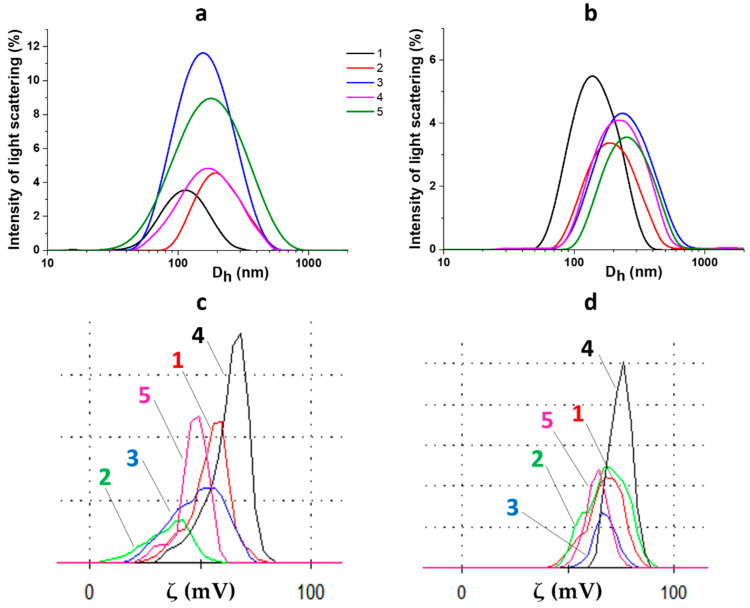
Particle size and zeta potential distributions for CSS/IO_3_^−^ systems before (**a**,**c**) and after (**b**,**d**) visible-light exposure for 1 h. 1, 2, 3, 4, and 5 correspond to systems 8, 10, 12, 14, and 16 (see Table, Figure 1B), respectively.

**Figure 6 gels-10-00809-f006:**
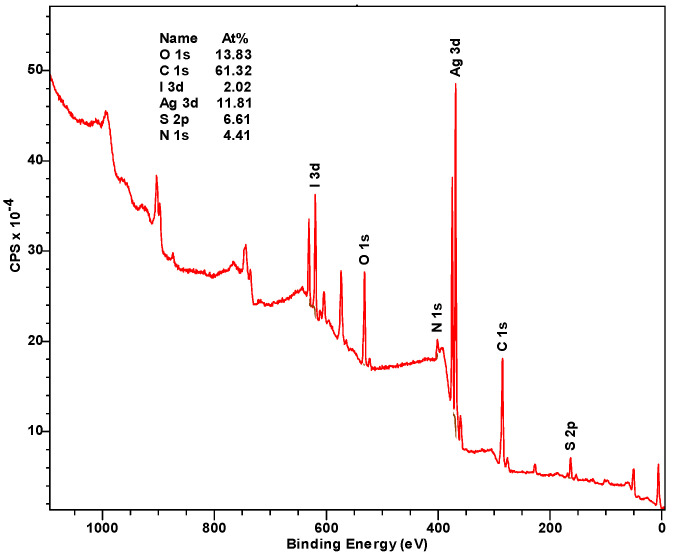
The XPS survey scan of CSS/IO_3_^−^ gel **16** (see Table, Figure 1B) after visible-light exposure for 1 h. The XPS spectrum for the non-irradiated hydrogel **16** remained unchanged.

**Figure 7 gels-10-00809-f007:**
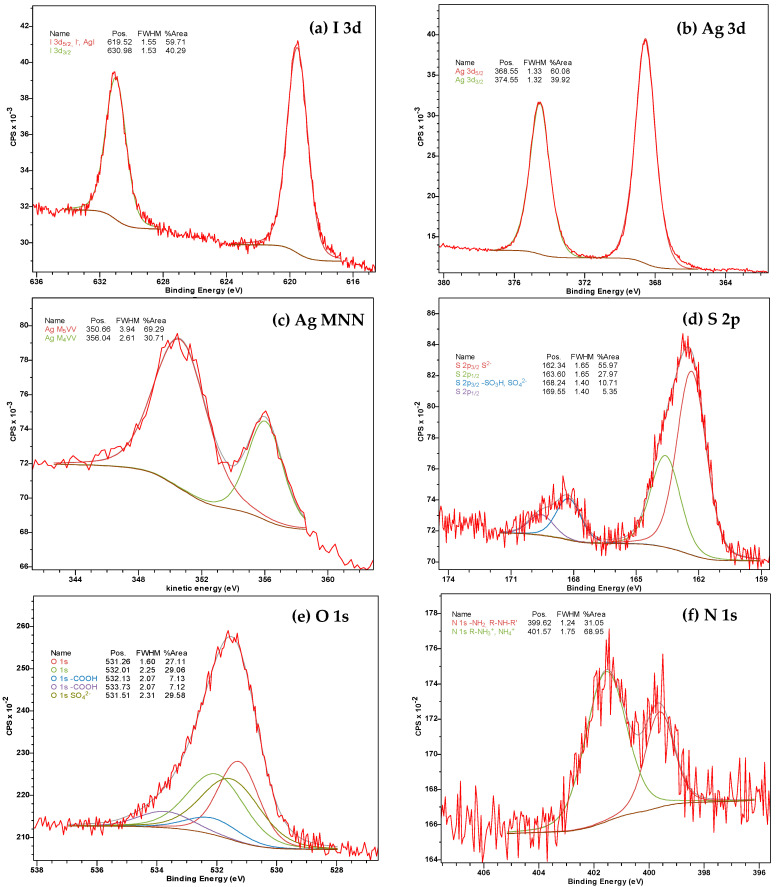
High-resolution XPS spectra of (**a**) I 3d, (**b**) Ag 3d, (**c**) Ag MNN, (**d**) S 2p, (**e**) O 1s, and (**f**) N 1s for the CSS/IO_3_^−^ gel **16** (see Table, Figure 1B) after visible-light exposure for 1 h.

**Figure 8 gels-10-00809-f008:**
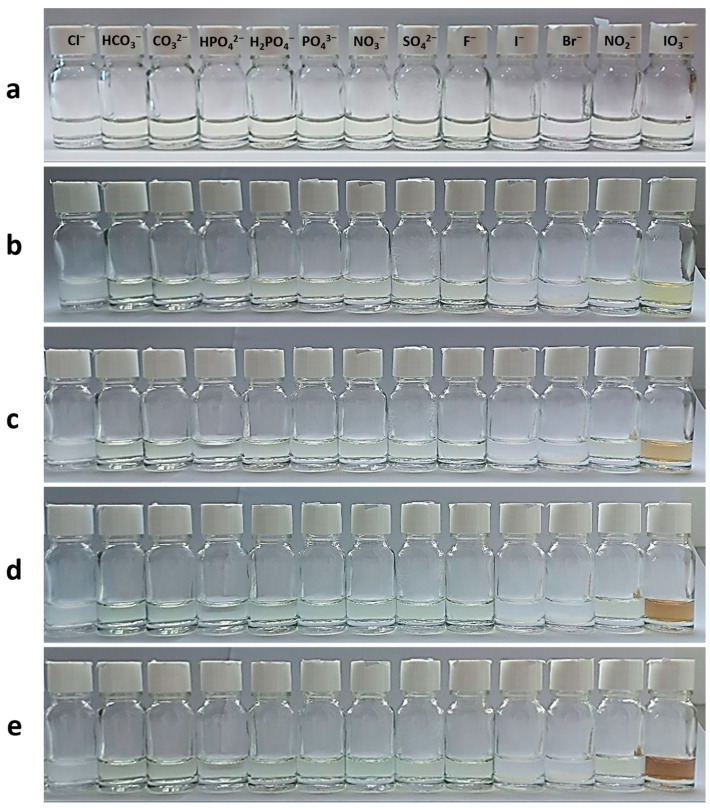
Photos of CSS with various anions: (**a**) immediately after adding electrolyte; (**b**) after 24 h in the dark; (**c**–**e**) after visible-light exposure of samples (photo (**b**)) for 20, 40, and 60 min, respectively. (**f**) the UV spectra of CSS with various anions (samples on the photo (**e**)) after visible-light exposure for 1 h. The final electrolyte concentration corresponds to the system 16 (see Table, Figure 1B).

**Table 1 gels-10-00809-t001:** Chemicals used in this study.

Chemicals	Purity	Producer
L-cysteine	>99%	“Acros” (Belgium)
Silver nitrate	>99%	“Lancaster” (UK)
Electrolytes: KClO_3_, KBrO_3_, KIO_3_, KCl, KBr, KI, KNO_2_, KNO_3_, K_2_SO_4_, K_2_CO_3_, KHCO_3_, K_3_PO_4_, KH_2_PO_4_, K_2_HPO_4_, NaF	pure	“Agate-Med” (Russia)

## Data Availability

All data and materials are available on request from the corresponding author. The data are not publicly available due to ongoing research using a part of the data.

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
