# Peer review of "L-Cysteine/Silver Nitrate/Iodate Anions System: Peculiarities of Supramolecular Gel Formation with and Without Visible-Light Exposure"

_gels, 2024, doi:10.3390/gels10120809_

Round 1

Reviewer 1 Report

Comments and Suggestions for Authors

Manuscript reports some photo responsive materials with several characterization method and very good discussion

- Lines 102-104 “To initiate the gelation process, it is necessary to lower the value of this charge by adding various low molecular weight anions to CSS [9,15,37,38]. Are there 4 necessary references for short sentence?

- In the figure 1A, you tried to represent “r.t.” for room temperature?, change r.t. for 25 °C or room temperature

- Figure 2c and 2d have numbers 1 to 5, but in the captions is no information about plott 1, 2, 3, 4, and 5. Include this information

- Figure 3, 3er column (next to light column) was difficult to see the information, improve it

- Manuscript has some typos, revise and correct them

- Line 352 “samples were irradiated by a visible light for 1 h” why selected 1 h for irradiation time

- Figure 3, why reported images dark and light

The manuscript has 8 Figures with 47 images or plots, there are many, try to summarize or move to supplementary materials

Author Response

Dear referee, thank you very much for such detailed analysis of our manuscript! All corrections are marked in green color.

1) Question: - Lines 102-104 “To initiate the gelation process, it is necessary to lower the value of this charge by adding various low molecular weight anions to CSS [9,15,37,38]. Are there 4 necessary references for short sentence?

Answer:  It has been corrected.

2) Question: - In the figure 1A, you tried to represent “r.t.” for room temperature?, change r.t. for 25 °C or room temperature

Answer: It has been changed for “room temperature”.

3) Question: - Figure 2c and 2d have numbers 1 to 5, but in the captions is no information about plott 1, 2, 3, 4, and 5. Include this information

Answer: Yes, thank you! The information has been included.

4) Question: Figure 3, 3er column (next to light column) was difficult to see the information, improve it

Answer: EDS results were obtained immediately in the form of images, rather than plotted graphs, and software cannot improve them.

5) Question: Manuscript has some typos, revise and correct them

Answer: Typos has been corrected.

6) Question: Line 352 “samples were irradiated by a visible light for 1 h” why selected 1 h for irradiation time

Answer: When we revealed this effect by only our eyes, it turned out, that after visible light exposure of samples for more than one hour, their color didn’t change significantly. Therefore, we started from this time. Certainly, in the future, we will study the kinetics of this reaction thoroughly and give an answer for how long it proceeds. 

7) Question: Figure 3, why reported images dark and light

Answer: Explanations are included in the caption to the figure.

8) Question: The manuscript has 8 Figures with 47 images or plots, there are many, try to summarize or move to supplementary materials

Answer: Thank you so much for this comment. We have analyzed various articles in this journal and believe that we have provided the necessary and sufficient amount of data without which the reader will not feel comfortable during acquaintance with the text of the main part of the paper.

Reviewer 2 Report

Comments and Suggestions for Authors

The submitted manuscript presents the results of the original research. This paper pertains to the identification of an atypical anion-photo-responsive gel system composed of L-cysteine, silver nitrate, and iodate anions. It has been demonstrated, for the first time, that iodate anions incorporated into CSS can function concurrently as a gelling and photosensitive agent in the visible wavelength range. Weak supramolecular gels were produced in the dark by mixing CSS and KIO3 at a millimolar concentration of anions.

This work is very interesting, well written and nicely presented. The Authors have well-documented their findings, all the sections are written correctly. The topic is also within the aim of the SI. However, I have also some questions and comments, presented below.

Line 19, why the authors keep using “cysteine-silver sol” and not “cysteine-silver solutions” as in i.e. 10.1007/s11172-022-3637-5 ? This work is not of mine, you don’t need to cite it.

Lines 95, 112, etc., it should be ClO3-, BrO3- and IO3- anions. I’m aware that the term “anions” indicates the negative charge, but the charge value (-1) should be stated as well.

Figure 1B, the number of decimals in the table should be consistent

Line 139, I’d rather say that to explain, at the molecular level, the observed results it would be highly recommended to include the molecular modelling methods, that can be used to not only explain the changes in the structure but also the reason for the changes in the UV-Vis spectra

Line 203, what do you mean by appropriate contents?

Figure 4, I’d recommend to create the additional graphs in which the authors would present the change of absorbance from time, of course at chosen wavelengths

As the mechanism of the observed effects is not known, it would be interesting to check if there’s an influence of the cation on the observed results. For example, if the authors have used the sodium salt (NaIO3) instead of potassium (KIO3), would the results be the same?

I guess the next step of this work would be to check the periodate and other XO4- ions, isn’t it? This can be mentioned in the conclusions.

The Authors mention in Line 93 that they suggest “a new approach for iodate anions detection” – it would be worth to investigate this topic even more. I.e., what would be the accuracy, LOQ, LOD, range of this method? Would would be its benefits when compared to other quantitative analytical methods?

Author Response

Dear referee, thank you very much for such detailed analysis of our manuscript! All corrections are marked in red color.

1) Question: Line 19, why the authors keep using “cysteine-silver sol” and not “cysteine-silver solutions” as in i.e. 10.1007/s11172-022-3637-5 ? This work is not of mine, you don’t need to cite it.

Answer: The concept of a “cysteine-silver sol”, not a “cysteine-silver solution”, was first introduced by me earlier to describe more clearly what we are dealing with. The concept of “solution” is confusing, because the reader may think that this is a true solution of two components, but this is a colloidal solution or sol. L-cysteine reacts with silver nitrate forming cysteine-silver nanoparticles.

2) Question: Lines 95, 112, etc., it should be ClO3-, BrO3- and IO3- anions. I’m aware that the term “anions” indicates the negative charge, but the charge value (-1) should be stated as well.

Answer: Yes, thank you. Everything has been corrected.

3) Question: Figure 1B, the number of decimals in the table should be consistent

Answer: Certainly, you are right! It has been corrected.

4) Question: Line 139, I’d rather say that to explain, at the molecular level, the observed results it would be highly recommended to include the molecular modelling methods, that can be used to not only explain the changes in the structure but also the reason for the changes in the UV-Vis spectra

Answer: Yes, of course, you are absolutely right! And thanks for the advice! In the future, we intend to simulate/calculate the UV spectrum of the system, but I don't think it will be so simple, everything depends on the basis in which the calculation will take place. However, based on the experimental results, we can generate the assumed structures and calculate the energies of HOMO-LUMO transitions, which are interrelated with the wavelength in the UV spectrum.

5) Question: Line 203, what do you mean by appropriate contents?

Answer: I am sorry, did you mean this one “Particle size distributions and zeta potential measurements for CSS/IO3- systems”? Here are graphs of particle size distribution and zeta potential distribution for the studied systems. Caption has been corrected! The measurements were carried out, certainly, for systems in a solution state, this is described in the experimental part, i.e. the initial gels were mechanically destroyed and diluted with water.

6) Question: Figure 4, I’d recommend to create the additional graphs in which the authors would present the change of absorbance from time, of course at chosen wavelengths

Answer: Yes, thank you, you read my minds! Of course, we will make it in the future using a conventional spectrophotometer! It is really interesting to see whether the absorption of the peak at 390 nm changes, which corresponds to the initial sol. The system is quite new and there is a lot to study here.

7) Question: As the mechanism of the observed effects is not known, it would be interesting to check if there’s an influence of the cation on the observed results. For example, if the authors have used the sodium salt (NaIO3) instead of potassium (KIO3), would the results be the same?

Answer: Thank you for this advice! I think that there will be no special changes in the mechanism of gelation, because we have investigated the effect of single-charged cations with the same anion to cysteine-silver sol, and vice versa. We have found out, anions influence the possibility of gel formation, while cations are also involved in the self-assembly process (especially when we moved to double- and three-charged cations), but they are less important. This is due to the positive charge of the surface of sol particles, as well as the anion's affinity to silver in the structure of sol particles.

8) Question: I guess the next step of this work would be to check the periodate and other XO4- ions, isn’t it? This can be mentioned in the conclusions.

Answer: Yes! I have added this information. Thank you!

9) Question: The Authors mention in Line 93 that they suggest “a new approach for iodate anions detection” – it would be worth to investigate this topic even more. I.e., what would be the accuracy, LOQ, LOD, range of this method? Would would be its benefits when compared to other quantitative analytical methods?

Answer: Yes, of course, there is a lot of work to be done in this direction. Firstly, it is necessary to check out the detection limit and whether it is possible to quantify this type of ions using the Bouguer-Beer-Lambert law, and in addition, in the presence of other anions! Thank you!

Round 2

Reviewer 2 Report

Comments and Suggestions for Authors

The Authors have revised their work, answered on all of my questions and comments. This manuscript can surely be accepted for publication.